# The Role of Leadership and Digital Transformation in Higher Education Students’ Work Engagement

**DOI:** 10.3390/ijerph20065124

**Published:** 2023-03-14

**Authors:** Valentin Niță, Ioana Guțu

**Affiliations:** Faculty of Economics and Business Administration, Alexandru Ioan Cuza University of Iasi, 11 Carol I Boulevard, 700506 Iasi, Romania; valnit@uaic.ro

**Keywords:** digital transformation, higher education, leadership, students’ engagement

## Abstract

Teaching and learning processes should be subject to continuous change due to the constant evolution of social, educational and technological environments, which ultimately results in higher levels of student engagement. The current paper describes the technological changes faced by higher education institutions as a result of digital transformation challenges. Further, transformational and transactional leadership styles’ effectiveness is regarded within the context of higher education institutions’ digital enhancements. Over time, these factors have led to contextual shifts that have disengaged students from learning and thus self-development. The current research aimed to examine how higher education institutions should apply different leadership styles within digitally transformed contexts so as to increase students’ learning engagement and reduce the risk of failure in their future developments within (inter)national labor markets. Data gathering and analysis involved a qualitative approach: an online survey was distributed, resulting in 856 responses. Through structural equation modeling, the data revealed a valid higher education digital transformation assessment tool; the results also emphasize the increased role of transactional leadership, as opposed to the traditional transformational style, within a highly digitized higher education institutional framework. Consequently, the linear relationship of students’ work engagement with leadership proved to also be enhanced by quadratic effects. The current study stresses the importance of internal and external peers in higher education performance through high levels of student learning (work) engagement through leadership and a uniformly developed digitally transformed higher education environment.

## 1. Foreword

As they have become active parts of international labor policies and social and political debates, digitalization and digital transformation are important contributors to individual and organizational higher education institutions across the world. In assessing the institutional digital transformation degree, one can use terms such as digital competencies [1], deep learning [2] or digital natives [3], all within an Industry 4.0 context [4,5]. Considering the strong effects on society and the reality of the need for further research and discussion, the potential changes imposed by digital transformation are likely to be disruptive changes, and the predicted effects need to be analyzed in relation to the large array of industries and specific contexts. Therefore, the current changes enhanced by higher education digitalization phenomena have led to increased research demand for educational sectors, thus ultimately resulting in industries’ and businesses’ need for change.

Despite the fact that, until recently, leadership has been overlooked, nowadays, it has acquired dominant power across the educational and business worlds, arguably becoming one of the most important topics specific to the social sciences. The evolutionary origins of leadership (generally used as a study variable for solving coordination and collective problems that include action and conflict) suggest converging ideas that support further development; thus, the digital age has been predominantly enforced with the help of the evolutionary roots of leadership. Moreover, leadership and followership share common properties, hence the prominent leadership styles known and practiced today under institutional umbrellas. Transformational and transactional leadership styles practiced under stable organizational environmental conditions have the capability of predicting the performance of those units. For the higher education system, the practice of different leadership styles (adopted by educational staff) leads to higher learners’ engagement; therefore, the teacher–student performance resides within a complex higher educational institutional context, where the implemented digital transformation instruments play a major role.

Despite the fact that students’ engagement has been subject to considerable research attention for the last two decades, it was not until recently that students’ involvement, experience and engagement were subject to research-led teaching [6,7]. With higher education institutions confronting narrowing economic conditions, attracting and retaining students by ensuring their development and workforce integration success matters more than ever. For this reason, the previous individual developments of students within higher education enrollment programs, along with the environmental conditions (digitally transformed, leadership-enhanced) of the institution where they study, contribute to their success during and after their time as a student. In cases where students’ engagement has the ability to deliver on its contextual promises, it can be converted into a magic wand, thus increasing the potential for further evolution in the workforce and/or entrepreneurial markets.

To close the current research gap, it is important to further clarify the specifics of terminology in regard to digital transformation, leadership and students’ engagement in order to obtain an in-depth overview of the educational and social fields and analyze how students react to (digital and leadership) changes. In sum, it is essential to identify the pre-existing competencies of students to determine which ones should be further developed so as to cope with the new digital environment, ultimately leading to higher learning engagement and easier insertion into the labor market.

However, to promote learners’ necessary competencies, thus resulting in an increase in their further development, a framework of an appropriate learning environment, course content, teaching methods and behaviors and digital media is required [8,9]. A didactic design enhanced by leadership tools and practice to confront challenges needs an appropriate content analysis in regard to students’ entry requirements. It is likely, therefore, that higher education institutions will be subject to a massive redesign of teaching and learning processes. For designing higher education digital transformation processes, it is crucial to determine the educational baggage that learners possess upon their enrollment; the teaching staff also needs to rely upon a clear educational status quo in regard to the needs and expectations of students, as well as past teacher–learner experiences. For designing their higher education process, students require adequately digitally equipped higher education institutions and teaching staff who will consider their needs based on active (reality-anchored) societal challenges, thus allowing them to fully engage in higher education processes [10]. In short, it is important to know the digital competencies, teaching environment history and students’ expectations for higher education institutions to develop adequate study programs and curricula. Therefore, the current research addresses the following research question: *on which students’ leadership and digital requirements can a higher education institution rely to develop study programs that would support students to become (learning) engaged and prepare them for higher education digital transformation?*

The growing digital transformation (DT) and its use within the higher education system have forced leadership to look toward achieving highly motivated and engaged learners so that their future operational efficiency and job satisfaction can be achieved. The vast majority of the research literature titles are focused on the business advantages and disadvantages of adopting and implementing DT, but very little of the available research literature has focused on the relationship between digital transformation, transformational and transactional leadership, and work engagement within higher education institution environments. Organizational DT drives management to prepare and motivate its students, but efforts must be directed toward exploring how higher education institutions manage the issues of leadership and students’ engagement under the DT umbrella.

To answer the research question, a uniform discussion basis for digitalization and digital transformation terminology is in order. Explicitly, it will enable the discussion of the effects enhanced by the digital framework in the higher education environment. Further, transformational and transactional enhancements of leadership regarding the digital transformation process will be discussed; the focus will also include integration within the abovementioned two-component process of students’ (work) engagement, as the ultimate result of digital leadership in higher educational processes.

The current project relies on the self-assessment of actively enrolled higher education students. To adapt to new institutional endeavors, it is essential to know students’ inputs in regard to teaching, digitalization and self-engagement specific to existing teaching–learning processes. By presenting empirical results, the authors present a connection to the theory, emphasizing the importance of adapting teaching and learning processes for students’ engagement with consideration of leadership styles and higher education institutions’ digital transformation.

As for the flow of the current research, first, the evolutionary perspective of the digitization-to-digital transformation was enhanced, followed by a succession of parallel perspectives of the main concepts—digital transformation, leadership and educational (work) engagement in the context of higher education—followed by the perspective of leadership and educational (work) engagement, discussed under a parallel literature umbrella. Methodological aspects include the use of the Smart PLS Software (v. 4. 0.0), where general and in-depth hypothesis testing revealed support for the initial assumptions of the current research, thus answering the research question. After a thorough discussion of the results, theoretical and practical developments are considered, followed by study limitations and final conclusions.

### 1.1. Transformation from Digitization to Digital 

Digitalization, also described by [11] as a binary number system development first acknowledged within the 17th century, has its origins within the Latin word digitus, suggesting an integer-based process, with clear and countable categories that keep their status of being countable and discrete in both value and time. The modern interpretation of digitalization suggests a discrete system conversion and integration process that uses binary data (with a minimum of two characters of 0 and/or 1) converted from analog into digital [5]. The process starts with streaming analog data, which are continuous and stepless, that are ultimately converted into digital (discrete information streams) data, with the information content being unaltered [12,13]. The current flow is also specific to processes and workflow situations. Digitalization is the term used to refer to the necessary steps in the conversion from analog to digital, thus transforming data into bits and bytes [14]. However, to realize digital transformation (DT), a more complex process of thinking and structuring is deemed to be integrated. Therefore, creative solutions may be explored for existing problems based on the use of existing technologies and the inclusion of available digital information within the process. The growing use and expansion of digital technologies, following an ever-increasing pattern, set the trajectory for fulfilling the goal of digital transformation, thus adapting existing processes by realigning the existing basic structures in accordance with fundamental business and/or technology changes. Increasing importance is stressed not on new arising technologies, but on solving resulting problems through new patterns of thought [15].

Concrete differentiation is needed for three terms: the initial digitization, digitalization and the resulting digital transformation. Each of the terms is able to build on the previous one. Digitization is only used when transforming data from analog into digital [16,17]. With the information newly available in a digital format, the workflows that might occur with the new data are also known as digitalization (thus improving the available feed), allowing the worker/user to access and use paperless data, thus facilitating the standard procedure. This process creates an increased potential for users to change and/or develop business models and activities through improved applications and the use of new technologies; the entire process of integrating digitized data and developing new applications and workflows, thus leading to new business models, is known as digital transformation. The terms’ clarification toward a common understanding is deemed necessary since various disciplines’ approaches led to inconsistency within the terms’ use, therefore allowing further clarification of higher education challenges triggered by digital transformation.

### 1.2. Educational Institutions and Digital Transformation

In the literature focused on work and society in the context of digital transformation [14,17,18], the subjects mainly concern structural interdependencies that include work processes, along with technology and the forms of organization; alas, people as peers in the workforce and the main users of such processes are relegated to the background and not considered for discussion. Although the literature recognizes the central importance of indicators specific to digital transformation [19], it is not the only objective worth mentioning. The process of digital transformation could not be developed without a skilled workforce; thus, addressing the question of how educational institutions are part of the process of training future employee generations for the emerging conditions created within the international labor market is deemed necessary [20].

Skill requirements specific to specialists in just one area of expertise will not be considered sufficient for the labor market, and thus, a complex range of qualifications will be required [21]. Market opportunities are developing with a trend for interdisciplinarity and increased arrays of personal competencies, along with a high degree of innovativeness and team spirit. Thus, educational institutions should enhance and foster an increased repertoire of educational competencies by focusing on teaching processes that foster interdisciplinarity and personal competency development. Due to the long process required by digital transformation, the discussion about the wide range of individual competencies has been a focus in the literature [22], but the value of digital competencies compared to knowledge-based skills has increased [23]. As the terms “digitization” and “digital transformation” have different uses (according to various disciplines), digital competencies also have a large array of definitions.

The term “digital competencies”, which is relevant to the use of modern technologies and information, represents only one of the relevant facets regarding the scientific discourse of individual competency. When referring to educational science [24], a better understanding of digital competencies [25] (Roll et al., 2021) can only be achieved by gaining a deep understanding of the meaning in regard to educational science competence. The results of a short literature analysis [1,26,27] show four main components that provide a uniform view in regard to competencies: first, a competency can only be assessed in specific situations (1), being drawn upon within particular cases (2), namely, in regard to a pre-defined subject (3), and may vary in relation to the degree of situational specificities (4). Subsequently, competencies may be expressed as a knowledge and skill overlap, to which both motivational and volitive components must be added, thus providing the age of digital transformation with the active involvement of individual/group digital competencies. According to the European Union (EU) Digital Competence Framework for Citizens (DigComp), the digital competence definition includes terms and expressions such as confidence, responsibility, engagement, information, data and media literacy, digital content creation, digital communication and collaboration, digital safety and intellectual property [27]. In addition to the extensive definition provided above for digital competencies, EC also first provides a deep understanding regarding the competencies themselves, which must include individual/team knowledge and attitudes, along with the individual skills possessed by an entity (human and/or institutional). Upon further analysis, one may notice that all the preceding components represent, in fact, the key elements that best define the concept of competence within an educational context, thus bridging the empirical theory of competence to its theoretical aspects.

In the context of higher education institutions, in the case of adopting and implementing such a competency-based approach, the need for adapting the development of teaching and learning processes to future developments is deemed to be brought to the surface in the form of education digitalization [28,29]. In the first view, the process of educational digital transformation implies the entire learning process, in accordance with the specificities of each generation, thus resulting in knowledge communication. Starting from a deep and complete understanding of the competencies within the education digitalization concept, the managerial process of higher educational institutions needs to adopt and implement preparation strategies and specific follow-up milestones, thus reaching a better understanding of how to address learning quality to meet society, industry and labor market requirements. Additionally, another question needs to be addressed, namely, how learning habits have changed in the context of the use of digital technologies [30,31,32]. Therefore, a new learner and teacher design arises, since their ability to access information simultaneously at any point in time and from different locations around the Globe by generating knowledge is of interest. New learning arrangements are created by new individual-specific learning settings designed to fulfill specific needs for the learning process [33,34]. Higher institutions need to address the issue of creating, for both leaders (teachers) and learners, on-site and online interaction platforms that may function in an alternative or separate mode [35,36].

### 1.3. Digital Transformation in the Context of Higher Education’s Traditional Activity

Digital transformation has shaped students’ evolution across secondary school education and continuing across the higher education framework, from both a self-studying and a teaching perspective; the insertion of online access to information, combined with remote communication technologies, shaped both teaching and learning behaviors, enriching the necessary skill array for surviving within a digital era. Higher education institutions’ digital transformation is a direct result of students’ subsequent insertion within the global employment system [37]; therefore, the current educational digitalization process requires assessing and adapting the respective processes to the global digital transformation trendline.

The question of adapting higher education institutions to the ongoing informational and communicational framework [38] includes three components. First, valuing the novelty of learning experiences through an increased number of institutional and private learning settings is regarded as being of utmost importance. Students and higher education representatives together rely on an (inter)national outcome-oriented labor market hallmark. Additionally, social aspects, such as private, institutional or legislative aspects, should also be considered for the undertaken educational digital competencies [18].

Further, one-size educational models are found to be undertaken by higher educational institutions [38], thus increasing the need to adapt and reorganize standard models to students’ needs in light of new labor market requirements and technical evolution. 

A serious problem in the emerging technological workforce context is related to the (dis)appearance of occupational profiles, thus creating the need for new strategies for creating and delivering educational content. Artificial intelligence (AI) already has the ability to augment workforce learning processes [39,40], providing future frameworks where the job taxonomy could be augmented, thus creating political behavior and new internal/external policy demands [41]. The main concern regarding future graduates in the context of the human–machine-learning work environment should stimulate higher education institutions to trigger the digital competency teaching process [42].

The emerging technological environmental expectations facing higher education systems compel the involved entities to overlook digital transformation in the form of innovations such as methodology or media [43] but to instead develop cyclical processes (PDCA) that both derive and create (graduate) workforce prospects through perpetually adapting teaching curricula [44].

The development of students’ digital competencies within a higher education framework creates a shift in the teaching focus from the historic–current students’ needs to the presently transforming teaching and working needs. For this reason, teaching–learning settings must also enhance digital-setting lecturers’ competencies [45]. These teaching prerequisites [46] specific to the higher education environment, from a holistic view, must also establish entry requirements for both lecturers and learners, thus supporting the continual redesign of digital competence development, preparing them for the increasing array of professional fields [47].

The previous literature argues that the focus in regard to the acquirement of digital competencies should not rely on digital natives [3] but primarily deal with lecturers’ development of digital competencies [38]. Despite the fact that different generations gain digital literacy through the use of everyday-life applications and digital devices, practical experience proved that mainly first-semester students do not have a native-digital ability to transfer and apply their technical skills to the teaching–study situational framework [48]; it appears that the heterogeneity of their digital skills derives from the large array of factors that might imply the use of digital transformation, such as social and private interests, along with the previous educational environment, thus influencing their adaptability to higher education institutional platforms, technology-supported educational opportunities and challenges.

It is important to highlight the fact that when measuring any kind of competency, researchers need to primarily pay attention to the fact that such measures can only be performed per se in the form of measuring and reporting performance (i.e., in action). Since such actions cannot be proved in a questionnaire, the research instrument was used to ask students to perform a self-assessment of their digital competency level.

The model regarding the digital competency assessment comprises six competency areas, starting with digital competency basics and initial access and continuing with information access and literacy in regard to digital data, followed by an assessment of digital communication and learner–learner/academic tutor digital collaboration, thus leading toward content creation and general and adaptive safety; the last component pertains to digital learner problem solving and further learning developments during online (as opposed to on-site) self-development.

**H1.** 
*Higher education digitalization is positively related to six attributes, namely, Attitude toward Internet—ATI; Equipment and Digital Service Usage—EQP; Online and Social Media Services—SMS; Digital Service Usage for Learning Outcomes—LDS; and Teaching and Learning Expectations—EXP.*


In order for higher education institutions to prepare curricula and reach an equilibrium in regard to first-year students’ digital transformation professional competence, it is deemed necessary to perform initial assessments. In Romania, the methodological system does not compel institutional entities to perform such studies; therefore, a pre-approved digital didactic methodology is not yet available. Despite the current situation, other EU countries, such as Germany and Switzerland, have developed and currently apply studies and programs in regard to higher education students’ digital competencies [49,50].

### 1.4. Digital Transformation and Leadership

From a teaching–learning perspective, the higher education digital transformation perspective requires a necessary organizational change that results in a digitally enabled institutional framework, validated across a large array of fields and departments. A legitimate institutional digital transformation must be adopted first by organizational members [51,52], and thus, a belief system needs to be enforced. From another perspective, the leadership of higher education teachers, mentors and lecturers resides within the organizational values, subject to continuous change over time [53]. For a higher education institution’s internal environment to become subject to digital transformation processes [54,55], the services performed should be included within an internal digital organizational culture [56]. Such changes are not deemed possible in the absence of leader and stakeholder interaction platforms, thus becoming part of agile institutions [57] that place focus on digital transformation through representation, content development and knowledge structures, bringing value to teaching–learning processes [58].

The organizational leadership roles that emerge during organizational culture shifts are common in the literature [59,60,61]. Institutions may achieve leadership success within the digital age by adopting different behaviors, often categorized as transformational and/or transactional. Transformational leadership inspires and intellectually stimulates followers, as opposed to transactional leaders [62], who actively apply contingent reinforcements to followers. By paying attention to emergent technologies, adapting investment strategies to digital trendlines and leading team changes efficiently [63], leaders create and support followers’ digital transformation mindsets [64,65], thus building institutional collaborative networks of leaders (teachers) and followers (students) and thus developing digital competencies and becoming digitally literate. Within the digital literature, transformational leadership is highlighted as a leadership style, since it enhances followers’ trust, morality and self-sacrifice for the benefit of highlighting the workgroup’s immediate needs and their satisfaction [66]. Therefore, transformational leadership within the digital transformation institutional environment becomes digital leadership, enhancing both technology and leadership features [67,68].

The institutional innovation absorption capability, according to the literature [69,70], is greatly enhanced by transformational leadership. Within the specific framework, a new leadership style is born, digital transformational leadership [63,71], thus realizing a positive influence on institutional innovation absorption performance.

The critical role that leadership has within the organizational information engagement framework [72] involves applying existing digital strategies to SMACIT technologies (Social, Mobile, Analytics, Cloud, Internet of Things), thus generating new value for both leaders and followers [73,74]. Institutions subject to digital transformation leadership create frameworks for internal unit digital meetings, an involvement that generally leads to increased strategic business knowledge and a higher freedom degree that enhance higher education institutions’ ability to adopt strategic decisions [75,76,77].

The institutional digital orchestra created and enhanced by digital transformation [78] is connected to the leadership’s ability to preserve the organizational objectives’ continuity [75,78]; from the initial theory point of view [79], this might be a transactional leader who aims for the institutional digital continuum and transition promotion in the pursuit of default objectives. The benefits from sliding from transformational to transactional digital leadership may be psychological (in the case when the leader–follower interaction may be enforced through direct appreciation or peer applause) or material (in the form of grade bonuses, advancement or an average grade rise). The transactional leadership component, i.e., contingent reward (CR), within a digital framework favors incentive creation for followers to fulfill their obligations [80,81]. Digital transformation involves strong leadership coordination, considering the idiosyncrasies of all organizational members and accessing both higher and lower organizational layers in regard to staff and followers’ mobilization in response to DT changes and challenges [30,82,83]. In the pursuit of contractual obligations, stakeholders must be involved in internal change processes to fulfill their contractual obligations, thus contributing to the higher education organization’s defined goal achievement [84,85]. Transactional digital leadership has undertaken the role of exchanging assurances [86,87], reaching agreeable compromises, negotiating organizational environment conditions and recognizing and rewarding satisfactory attempts. From this point of view, by acquiring digital transactional leadership behaviors, higher education institutions are susceptible to reaching a vantage point.

Staff autonomy and democracy within higher education institutions can lead to a breach of the old autonomous leadership systems and favor reaching a compromise with DT. Organizations that employ such an administration method and monitor step by step the success and failures created by the perceived change within both leaders and followers subscribe to management-by-exception active (MBEA).

For the development of the current research, the one component of transactional leadership and the two-element shell of the transactional leadership style (contingent reward (CR) and management-by-exception active (MBEA)) have been assessed through the critical eye of the student–teacher academic relationship within the undertaken academic program as observed by both parties. Of course, the academic behavioral language has deepened within organizational customs and culture, which is the reason that a self-assessment of the follower (student)–leader (teacher) behavioral (spider) web was deemed necessary.

By undertaking critical decisions when performance indicators indicate it is appropriate, practicians of such a leadership style elicit democratic cross-functional cooperation from the participants involved, thus resulting in a decentralized process of decision making regarding DT procedures [83]. For this reason, the following hypotheses arise:

**H2a.** 
*Transformational leadership is positively related to digital transformation.*


**H2b.** 
*Transactional leadership is positively related to digital transformation.*


### 1.5. Digital Transformation and Educational (Work) Engagement

Work engagement has been perceived for a long time as an organizational challenge. By adapting higher education institutions to a digital environment and introducing lecturers and learners within a new DT architecture, the problem of work engagement rose to a new level, since most human resources representatives were left behind. Workforce/student learning engagement is considered to be an aggregate of students’ learning experiences within the higher education institution they belong to [88]. Therefore, in the current study, when referring to students’ work engagement, the authors intend to assess the involvement (as engagement) of each student within their personal higher education academic trajectory; depending on the specifics of the academic field and teaching–learning requirements (as solely theoretical and/or mixed with practical study requirements—such as seminars and/or laboratory or practice stages), the levels of academic self-involvement widely vary. In the current case, the specifics of the fifteen faculties involved (with a large variety of study fields) account for a variety of teaching and learning processes that will ultimately be presented and assessed within the current study as student work engagement, necessary for becoming graduates. A slight difference might arise when considering the possibility for work and learning to be juxtaposed in the given environment; for this reason, it is important to clarify that students’ work engagement is commonly referenced so as to bind them by the book to their mandatory academic requirements, while learning engagement also takes into consideration additional implications of extracurricular practice, training, academic writing and more. However, such a difference is difficult to measure, especially due to the fact that universities do not provide a clear academic collaboration framework with the public/private environment, but in many areas, student practice within these environments is graded; thus, for the accuracy of the current study, to avoid this bias, learning and working students’ engagement is singularly identified. Moreover, new generations whose daily activities are anchored within social media face the challenge of consistent educational engagement within higher education institutions, generally within teaching–learning frameworks that are very traditional or too advanced for the ordinary student to be able to simply engage with the daily curriculum.

For this reason, collaborative organizational DT frameworks [89] that rely on digital ecosystems that engage multiple layers of higher institution members trigger a forward step within the organizational culture. DT relies on transformational business and organizational activities specific to higher education institutions by accelerating the impact of digital change across specific internal and external environments. There are a large number of studies that explore the role of DT in the education world [90,91], mostly introducing users to social media platforms, the Internet of Things (IoT), the Internet of Everything (IoE), cloud storage and technologies, and mobile data analytics [92,93,94]. Students’ engagement with technology-enhanced learning activities plays an active role in creating value and facilitating new experiences for both teachers and learners. If considering digital leaders’ opinions within the digital engagement framework, higher education institutions are losing business opportunities (i.e., students’ effectiveness) in the absence of DL-driven educational processes [95].

The current educational DT climate stresses the importance of educational (student) engagement within an environment where a transformation toward transactional digital leadership has arisen, since digital transactional leaders stress the importance of employee fulfillment and goal achievement [96]. The new leadership shift has increased the complexity of the higher education environment, since most of its representatives focus on the learner’s engagement, therefore striving to create accessible platforms that provide the necessary support for students and academics together. The higher education business model has therefore been altered forever by the evolution and implementation of DT; however, human interaction will never be susceptible to being fully replaced by technological advancement; therefore, the reasoning behind which higher education institutions listen and respond to learners’ needs and requirements has been transformed, with reports showing better results compared to the pre-DT era [97].

In light of previous results in regard to employee engagement through DT [98], the specific teacher–student working space has been altered by the introduction of artificial intelligence and breakthroughs specific to the newly adapted virtual environment. Despite the major improvements granted by institutional DT, traditional learning attitudes are shifting, and the pressure created by job disappearance and the required task-solving speed encourages future generations of employees to rise to the technological challenges that enable businesses around the world. As higher education institutions’ DT phenomena advance, the advantages of engaged student generations are highlighted [99,100]. The following hypothesis arises:

**H3.** 
*Digital transformation is positively related to educational (student) engagement.*


### 1.6. Leadership and Educational (Work) Engagement

The literature does not provide information regarding the traditional relationship that involves leadership and students’ learning engagement; therefore, the current assumption has not been empirically tested or studied. Despite the current situation, researchers seem to focus their attention on the large effect that leadership has on generally defined work engagement, often presented as positive and significant [101]. With the milestones that characterize a higher education organizational change due to DT, the importance of organizational climate functioning under the leadership umbrella has a direct effect on the particular results of the change; the process includes followers’ educational engagement, commitment, perceptions and involvement, highly leveraging the process benefits [102]. As previously mentioned, digital transactional leadership has been steadily preferred compared to the transformational style; contrary to the current observation, leadership is an excellent driver of innovative and positive communication endeavors specific to higher education peers [103]. Previous authors claim that transformational leadership is a positive openness to change—a work engagement moderator [104,105]. The desired outcomes within a higher education institutional DT process are highly dependent on the leader’s abilities to stimulate and reward followers; hence, transactional leadership highly affects learners’ development and training [106,107,108]. Prior studies on leadership result in increased work engagement and performance [109], while innovation and self-involvement within the daily routine are nearly absent.

The construct that defines employee engagement as individual involvement within daily organizational tasks [110] is nurtured and developed by leadership practices able to relate to individual psychological aspects that ultimately result in self-efficacy and goal achievement [111].

By creating a feeling of belonging [112] and individual satisfaction [113] and reducing absenteeism [114], leadership has an undeniable role in improving work performance due to the open framework created by the institutional architecture involving management and followers [94]. Moreover, leadership not only benefits teachers’ and students’ work engagement but stimulates and improves the overall operational quality specific to a higher education institution. Considering the various studies’ explanations above, the following hypothesis arises:

**H4.** 
*Leadership is positively related to educational (work) engagement. Moreover, no studies have conceptualized the actively enhanced connection that includes higher-education-enrolled students’ learning (self-working) and academic leadership under the evolutionary umbrella of higher education digital transformation. Similar research in regard to solely academic leadership style(s), students’ engagement and/or the digital transformation of universities are fleeting within the literature, but an enhancement study attempt including the three concepts together has not been revealed so far within the published academic literature. For this reason, an in-depth analysis of the evolutionary content of academic digital infrastructure was deemed necessary, along with clear characterizations of all the abovementioned concepts.*


## 2. Materials and Methods

According to the theoretical methodological procedure for performing a survey analysis, a pilot study is strongly recommended for pretesting the used research instrument, a questionnaire. To conduct the current research, a Google Forms survey link was posted online on a large array of online platforms. Twenty-six answers were gathered and deemed ready to be analyzed and interpreted. The previous literature [115,116,117,118] suggests that in order for the researchers to prevent and/or avoid impairments and implement changes considering the used methodology, survey administration, data gathering, data interpretation, launching and assessment, a pilot study is highly recommended. In the current study, the pilot study results revealed the fact that no misunderstandings were indicated and the items did not present miswording; therefore, the research instrument was considered to be relevant for the development of further statistical analysis.

A total of 856 participants, aged between 18 and 35+ years old, agreed to take part in the current study. There was an uneven split, as 82% belonged to the 18–23-year-old category (720 respondents), and 10% were aged between 24 and 29 years old, while the percentage increased (5.8%) for the 35+-year-old segment; the least representative age segment is 30–35, with only 1.7%. As for the data on the respondents’ gender, the data show that 37.7% were male and 62.6% were female. The data used for the development of the current study were gathered from a single university with 15 active faculties. Since student programs in different faculties have different lengths and effects, for the development of the current manuscript, the decision to not consider an even number of respondents for each faculty was agreed to. Therefore, the largest groups of respondents belonged to the Faculty of Geography and Geology (15.1%), Faculty of Business Administration (12.5%) and Faculty of Orthodox Theology (9.4%), while the fewest responding students were enrolled in the Faculty of History (2.2%) and Faculty of Physics (2%).

### 2.1. Measures

The current study’s measures were from a 57-item questionnaire that was distributed via the Google Forms online platform. The research instrument includes three domains of study: higher education digitalization, leadership and students’ work engagement. For the dimensions exploring whether leadership is transformational (TL) or transactional, i.e., contingent reward (CR) and MBE active (MBEA), the questionnaire was reduced in scale by item reconfiguration. The reasoning behind this choice is the increased length of the research instrument that, if not narrowed down, would have favored a pattern of misuse of the proposed scale; to safeguard against blank responses, the “Required answer” setting was also used.

The research instrument design includes three parts: first, the demographic status section required the Field of Studies, Age and Gender of the respondents; further, a second part explored higher education digitalization by using five latent variables consisting of 27 items; the third part that was developed by considering the higher education leadership encompasses transformational leadership’s latent variable, consisting of 5 items, along with the transactional leadership components, namely, contingent reward and MBE active, each consisting of 3 items. The last part pertains to students’ engagement, consisting of three latent variables: Autonomy (AUT) and Social Support, (SS) each including three items, and Educational Activity Involvement (EAI), consisting of eight items. Instead of a 5-point scale, a 7-point Likert scale was used; this decision’s reasoning is based on the specialized literature showing that data accuracy increases when using a scale of 7 points or more, thus decreasing for scales of 5 points and below. For this reason, a 7-point Likert scale was considered to be suitable for the current research, given online usage and sharing; within the given conditions, a symmetric 7-point Likert scale interface is considered by respondents to be more friendly for online completion and provides an increased number of selection options, resulting in the increased accuracy of the expressed options. It is worth mentioning that the realization of the set objectives by appealing to respondents’ individual reasoning faculties has been considered by the used methodology. 

### 2.2. Higher Education Digitalization Measures

The current variable consists of six latent variables (Attitude toward Internet—ATI; Equipment and Digital Service Usage—EQP; Online and Social Media Services—SMS; Digital Service Usage for Learning Outcomes—LDS; Teaching and Learning Expectations—EXP), where ATI, SMS and EXP consist of 6 items, EQP comprises 4 items and LDS involves 5 items; a 7-point Likert scale was used, with answers ranging from 1 to 7 (from totally disagree to totally agree).

The higher education digitalization measure is based on the questionnaire developed by the Austrian DigComp 2.2 AT Competence Model [17,27].

The initial internal consistency (IC) values for the involved variables were 0.67 for ATI, 0.45 for EQP, 0.72 for SMS, 0.65 for the LDS dimension and 0.44 for EXP. The IC can be improved by performing a statistical analysis and removing a number of items and variables.

### 2.3. Transformational Leadership Measures

The latent variable for transformational leadership derives from the four initially proposed dimensions [119] known in the literature as Idealized Influence (Attributes and Behaviors), Inspirational Motivation, Intellectual Stimulation and Individual Consideration, originating from the initial 45 items of the MLQ (the 5 X form) questionnaire. For the development of the current research, the instrument was adapted from the original 5-point Likert scale to a 7-point Likert scale, with answers ranging from 1 to 7 (from totally disagree to totally agree). The internal consistency of TL is 0.87.

### 2.4. Transactional Leadership Measures 

Six items were adapted from the original MLQ 5x form (Avolio et al., 1995), grouped within two dimensions, contingency reward and MBE active, each consisting of three items. A 7-point Likert rating scale was used, with answers ranging from 1 to 7 (from totally disagree to totally agree). The average internal consistency for CR had a value of 0.82, with MBEA averaging 0.77.

### 2.5. Student Engagement Measures

For the study of students’ work engagement, the authors used the AUT and SS variables, each consisting of three items [120], while the EAI variable comprises eight variables. Educational Activity Involvement is derived from the 9-item Utrecht Work Engagement Scale [121,122] and was adapted from a 5-point scale to a 7-point Likert scale. The average internal consistency for AUT is 0.82, and SS equals 0.81, while EAI averages 0.91.

### 2.6. The Analysis Strategy 

For the five constructs’ and eight subconstructs’ assessment and strategic development, the SmartPLS (v.4.0.0) software was used. The first step was to initially test the newly proposed research instrument (higher education digitalization) and assess its internal consistency and reliability. After construct validation, the analysis included an SEM analysis, followed by a Confirmatory Tetrad Analysis, and an Importance–Performance Map analysis was also developed; the analysis also included the study of curvilinear quadratic effects and specific indirect effect sizes and ended with a FIMIX segment analysis. The objective of the current research was to provide a complete assessment of the relation among higher education digitalization, leadership and students’ (work) engagement in the context of Alexandru Ioan Cuza University of Iasi’s organizational environment.

The choice of the specific software resides within the fact that by using a SMART PLS SEM analysis, the results would provide the authors with a better understanding of the advanced existing relations among the specific studied variables [123].

The SmartPLS software is recommended [124,125] for cases where the model includes at least one formative construct. It uses a partial least-squares algorithm that results in two models: the outer model is recommended when analyzing data in regard to the observable variables’ infrastructure and yields the latent variables; the second model (the inner model) provides the structural model residing within the proposed latent variables to produce different latent variables.

In the current research, the outer model was analyzed first, thus revealing the latent variables’ validity and reliability. Subsequently, when testing the inner model, the data revealed path coefficient values regarding connections among the outer model constructs [126].

### 2.7. Method Setting and Sample

Based on the practice and pragmatism specified in prior literature findings, in the current study, a convenience sampling method that relies on voluntary responses was used. The data were gathered with the help of the online Google Forms platform, generating a hyperlink that was shared with various formal and informal educational communication platforms. To reach the respondents, a number of (faculty and private) online applications and electronic communication platforms were used. The research instrument design included the General Data Protection Regulation (GDPR), which was provided when first following the link to the questionnaire; the respondents were therefore guaranteed the lack of a requirement for and/or use of personal data, along with compliance with the strict confidentiality of the provided items’ selection. Respondents were also assured that the data would only be used for the purpose of academic research.

With the use of non-random sampling, a pilot test was conducted first, followed by hypothesis generation. As previously discussed, the current research included a convenience sampling method based on voluntary responses [127]. The survey provided the identification and definition of variables to the target audience, consisting of enrolled students within one of the fifteen faculties of the Alexandru Ioan Cuza University of Iasi. As the methodological provisions state, data from students enrolled in other universities or individuals currently not enrolled in one of the abovementioned academic environments were not considered valid and were therefore excluded from the current analysis.

The intended questionnaire design first provides a deeper understanding of the amplitude and limits of the digitalization concept by developing a five-variable research instrument as previously designed [17]. Furthermore, the research design is intended to assess whether there is a connection between higher education digitalization and leadership (one transformational variable and two transactional variables). A third link with the students’ (work) engagement was also considered by adding a three-variable component that comprises AUT, SS and EAI.

By agreeing to fill in the survey, the respondents agreed that no compensation would be provided. The survey was distributed online within a two-month interval (November–December 2022), resulting in a total of 856 responses; therefore, the average response rate was considered to be low.

## 3. Results

In order to assess a new instrument, according to the literature, a series of actions are in order [123]. The first recommended step is checking the constructs’ collinearity; by running a consistent PLS SEM algorithm, the standardized paths and Variance Inflation Factors were provided, generally used for characterizing the collinearity degree of model indicators. There are different thresholds for VIF values, ranging from below 5 to 10 [128,129]. For the use of the SmartPLS Software [130], the VIF values must meet the threshold conditions of being lower than 4 (<4.0); the VIF values for the current model are below the considered value, and therefore, no multicollinearity issues are identified.

The analysis continues with the performance of another consistent PLS SEM algorithm for standardized factors in order to determine the convergent validity of the proposed construct, for which the Average Variance Extracted (AVE) values were considered; the AVE value must comply with the threshold of an average >0.5 [131,132]; in cases that do not satisfy this condition, indicators with outer loadings <0.4 are to be dropped, while indicators with outer loadings in the 0.4–0.7 interval can be retained by the researchers if the decision does not affect the values of CR and AVE [133].

As the data from Table 1 show problematic AVE values for five variables (EQP = 0.369; EXP = 0.469; higher education digitalization = 0.236; LDS = 0.42; and SMS = 0.434), according to the literature criteria [134], for the measurement of convergent validity, the composite reliability values must be used in order to determine whether the measured item has sufficient convergence. According to the rho_c values, all variables have sufficient convergence, except EQP, whose rho_c = 0.698 (<0.7); therefore, the latent variable must be removed.

In order to satisfy the numerous literature criteria, since the higher education digitalization construct was not previously subject to peer review and assessment, the authors decided to report the values of Cronbach’s Alpha and composite reliability; as suggested, averages >0.7 are considered to be good for both of the considered indicators. For the current model, Cronbach‘s Alpha falls in a value interval of 0.44–0.93, while CR values are within the 0.69–0.93 interval.

From a simple analysis of the factor loadings, ATI3 = −0.002, ATI7 = −0.003, ATI1 = 0.433 and ATI8 = 0.345 are eliminated, leading the ATI AVE value to increase from 0.368 to 0.524.

Considering Cronbach’s Alpha, composite reliability and AVE values, the decision to eliminate the EQP variable was considered.

For the use of the SmartPLS software, the literature recommends that, when performing the instrument’s reliability analysis, instead of reporting Cronbach’s Alpha values, Rho_a values should be considered [135,136,137]. As the data from Table 1 show, the Rho_a interval 0.81–0.93 confirms the given construct to be a reliable composite.

The new construct reliability and validity values are presented below.

The results show that the model became satisfactory with sufficient convergent validity.

A new Smart PLS SEM analysis for standardized factors was performed, providing new path coefficients for each latent variable.

The standardized path coefficients, after removing a series of items from the current model, range between 0.1 and 0.6, adopting absolute values of <|1|, thus proving the idea that the model does not present multicollinearity issues (a case in which the path coefficient values would have been greater than |1|) (see Table 2).

For the assessment of the structural model, the literature suggests performing a bootstrapping procedure for assessing the path coefficients’ significance and the R^2^ values [138].

As the literature suggests, R^2^ values averaging >0.67 are considered to be substantial, those >0.33 are moderate, and those >0.19 are weak [139,140]. Other authors have agreed upon another R^2^ value classification that is considered valid independently of the applied field of study [141,142,143]; according to this new classification, weak correlations are within the 0.00–0.29 range, while the ranges for low (0.3–0.49), moderate (0.5–0.69), strong (0.7–0.89) and very strong (0.9–1.00) are considered to be more tolerant. Another author proposes a behavioral science classification of the effect size “r”, considered to be small when r = 0.10, medium when r = 0.3 and large for values of r = 0.5 or more [144]. An extensive interpretation of the explained variance for particular constructs requires R^2^ to take values ≥ 1 in order to be considered adequate. For the Smart PLS software, an R^2^ value below <0.25 is considered to be very weak, 0.25 ≤ R^2^ is considered weak, 0.5 ≤ R^2^ < 0.75 is moderate, and R^2^ ≥ 0.75 is substantial [145,146,147,148]. For the studied construct (see Table 3), the R^2^ values for students’ engagement are substantial, while all the other R^2^ values are considered to be weak or very weak.

For the model fit, the Standardized Root Mean Square Residual (SRMR) values need to be considered; for the current model, the Saturated model’s SRMR = 0.10, while the Estimated model’s SRMR = 0.15, thus falling in the acceptable value range of <0.1 [149] but not in the more conservative version of <0.08 [150]. Therefore, the model provides a good fit, and it does not present misspecifications.

To complete the design of the measurement model and determine the formative or reflective nature of the model indicators, by performing the Confirmatory Tetrad Analysis (CTA-PLS), the default reflective design of the current model was considered suitable for testing. The CTA-PLS analysis can only be performed for latent variables with four associated indicators or more; therefore, for the current model, except for CR, MBEA, SS and AUT, all variables were tested. The analysis of the *p*-values indicates a formative model for *p*-values <0.05 (the threshold for significant values) and a reflective model for *p*-values >0.05. Additionally, according to the literature, for variables with a large item number, a large threshold of 80% of the *p*-values was introduced, according to which if 80% of the *p*-vales are considered to be significant, then it is formative, while for an 80% threshold of insignificant *p*-values, it is considered to be reflective. After performing a CTA-PLS analysis with 10k subsamples and parallel processing with a 0.05 significance level for the current model, the results indicate that higher education digitalization, LDS and SMS are considered to be formative.

Further, the authors tested the inner model to uncover unobserved heterogeneity by performing a FIMIX analysis (FIMIX-PLS); the latent class method is considered to be best suited for estimating the probabilities of existing hidden segments [151] by using the estimates’ path coefficients (in the case of their proven existence). As the literature suggests, empirical research is often subject to hidden heterogeneity, since its external resources cannot be controlled or a priori prevented; for this reason, the latent class techniques proposed by FIMIX-PLS are suitable to be applied [151,152,153]. Further, the literature suggests the evaluation of unobserved heterogeneity as a routine technique for further analyses.

Assuming an 80% power level with a 0.15 effect size [154,155], for the size of the database, the analysis indicates a maximum number of 13 segments to be extracted.

In order to study the degree of segment separation (through the normed entropy statistic—EN) and the minimum values of criteria such as Akaike’s information criterion (AIC) modified with factor 3 and consistent AIC, further known as CAIC [123,156], the authors determined the number of segments to be retained (see Table 4). According to the literature, EN indicates the reliability of a partition; therefore, an EN value ranging from 0 to 1 indicates a higher-quality partition when acquiring a higher value [157]. According to another view, an EN > 0.5 allows the cutting of data for a predetermined segment number [158].

The initial results do not indicate a clear segmentation scenario; the literature analysis suggests that in cases where AIC 3 and CAIC indicate the same segment number, it is likely that the database will present the corresponding segment number; within the given case, AIC 3 indicates a three-segment solution (see Table 5), while CAIC indicates a two-segment solution [159].

After performing a discrete segment assignment, the data show no values exceeding the >0.2 threshold; therefore, the data do not have any segments, and thus, the unobserved data heterogeneity provides sufficient evidence for the results’ robustness [154,160]. In order to analyze which variables are good predictors and how well those latent variables perform, a further Importance–Performance Map analysis was performed. On the x-axis, we have the predictive power, or the relative influence that the predictor has on the outcome target variable, meaning that the performance is on the y-axis, that is, how well we measure the predictor. The higher on the y-axis and the higher on the x-axis, the better.

For the higher education digitalization variable, SMS and LDS are the most important predictors and perform the best. ATI has very little effect on the higher education digitalization variable, and therefore, from a practical standpoint, if there were resources put into it, the focus should be on SMS and LDS, as they should have the most impact on influencing higher education (see Figure 1). 

For the students’ engagement variable, BBEA is the most important predictor by far (therefore, a transactional leadership component), followed by SS and AUT. All the remaining variables are weak predictors with undistinguishable importance. For this reason, from a practical standpoint, if a higher education institution were to allocate resources for the increase in enrolled students’ engagement, the focus should be on EAI (see Figure 2).

For the transformational leadership variable, the most important predictor from the entire proposed construct is higher education digitalization through its components SMS, LDS, EXP and ATI; the least important predictors are CR and MBEA, which are specific to transactional leadership, enforcing once more the importance differentiation of the two included leadership styles (see Figure 3). 

The quadratic effect allows one to explore beyond the linear relationship to a curvilinear relationship, specifically a quadratic, which is the most common form of a curvilinear relationship. We tested the relationship between higher education digitalization as a predictor and TL, CR, MBEA and students’ engagement as separate outcomes. Moreover, the relationships between AUT, SS and EAI as predictors and students’ engagement as the outcome were also considered, thus estimating both the linear and quadratic coefficients. The higher education digitalization model’s curvilinear quadratic effects are statistically significant at the 0.05 Alpha Level only for higher education digitalization–TL and SS–students’ engagement relationships (see Figure 4); the model presents the path coefficients, *p*-values and R^2^ values specific to the inner and outer models.

For example, in the equation for the higher education digitalization–TL quadratic relationship, we used the following formula: Y = 0.134x^2^ + (−784)x, where the X^2^ coefficient represents the curvilinear quadratic effect, and the X coefficient is Beta.

We used Google Plot and treated the plot as a relative shape by using standardized coefficients.

There is an inverse curvilinear relationship (an inverted peak) between higher education digitalization and TL; as higher education digitalization decreases, there is less TL; after reaching a certain inverted peak, an increase in higher education digitalization does not affect TL by the same amount (see Figure 5).

There is a curvilinear relationship between SS and students’ engagement. As SS initially increases, there is more engagement from the students; it reaches a certain peak, after which more SS does not increase students’ engagement by the same amount as occurred previously (see Figure 6). 

Further, the estimation and analysis of the specific indirect effect sizes were performed. The analysis may be performed only for models with reflective factors.

If the *p*-value is too high (*p* > 0.05), meaning too much error, then it is rejected, but if the *p*-value is significant (*p* < 0.05), then the indirect effect hypothesis is supported if there is an effect observed [161]

For the current analysis, there are three cases in which a small effect is observed, namely, EXP → higher education digitalization → TL, ATI → higher education digitalization → TL, and LDS → higher education digitalization → TL, with *p* = 0 (thus below the threshold of *p* < 0.05), proving that the hypotheses are supported, with an effect observed (see Table 6).

If there is less than a small effect, but still less than 0, then it is recommended to take context into consideration and determine whether there is a small sample size or an underdeveloped area of research; if so, as long as it is statistically significant (*p* < 0.05) and the effect is greater than 0, we would say that it is probably supported, or at least provides useful information as a signal for future work. In the current case, there are five indirect effect hypotheses that apply to the current direction (SMS → higher education digitalization → TL, LDS → higher education digitalization → students’ engagement, ATI → higher education digitalization → students’ engagement, EXP → higher education digitalization → students’ engagement, SMS → higher education digitalization → students’ engagement).

For the rest of the hypotheses, even though the sample size is n > 400, [162], since the current analysis refers to an underdeveloped research area, the rest of the hypotheses referring to the existence of an indirect effect are rejected.

### Hypothesis Testing

According to the first hypothesis, higher education digitalization is positively related to the six attributes ATI, EQP, SMS, LDS and EXP. We tested the model in order to analyze the path coefficients and determine whether the first hypothesis is entirely validated; according to the factor loadings, the relation between higher education digitalization (HED) and SMS has the highest value of 0.368, followed by LDS, EXP and ATI with values above 0.23, while the EQP loadings displayed the lowest value (0.176). After a thorough statistical analysis, the authors decided to exclude the latent EQP variable, therefore partially supporting H1.

The authors further tested whether there is a connection between higher education digitalization and three leadership components, namely, transformational leadership (TL) and two transactional leadership variables, CR and MBEA. According to the factor loadings, all the relations are positive, therefore supporting H2a and H2b.

Further, the relationship characterizing higher education digitalization and students’ engagement was tested. The data show a positive connection; out of the three student work engagement components, EAI proved to have the strongest effect on the digitalization component of the higher education process. Thus, the data support H3.

The last hypothesis was tested to prove an existing relation between TL and students’ engagement. After studying the quadratic effects among most of the variables, the results show an existing inverse peak relationship, thus supporting H4.

## 4. Discussion

According to the fourfold aim of the current study, the authors started with the proposition and validation of a new assessment tool for the higher education digitalization concept. Further, leadership in two of its forms, transformational (TL) and transactional (CR and MBEA), was explained by higher education digitalization within an institutional context. Following the current analysis, the researchers tested whether there is a connection between higher education digitalization and students’ involvement in the study process, and the connection was found to be positive. Ultimately, beyond linear effects, as the fourth aim, the authors took into consideration the possibility of quadratic (thus non-linear) effects that might characterize the construct’s connections; the results showed positive results for the component of students’ work engagement with TL and SS.

The importance of the current study resides within the unique approach to studying higher education digitalization within a single institutional environment. The qualitative assessment tool used was applied considering not only the institutional leadership and/or digital transformation approach but also the student perspective, studied in the context of their (work) involvement within their daily study tasks. In terms of the current approach, we will further discuss the findings at length.

### 4.1. The Six Attributes of Higher Education Digitalization

By departing from previous theories [163], the contribution of the six attributes to the development of the institutional digital transformation must be tested in accordance with composite reliability and variance values; in the current case, as previously mentioned, only five of the six attributes were confirmed. Even so, the tool’s utility within the higher education institutional environment was confirmed, providing initial data for further digital transformation developments and/or organizational improvements. Moreover, the development of a supporting study should be pursued to determine whether there are any correlations among higher education digitalization and managerial attitudes and practices, traditions and customs, internal and external technology investments, and individual and/or organizational performance. Within the framework of performing such a study, the proposed higher education digitalization assessment tool could find importance and receive interest within individual and/or conglomerate (ministerial/business groups) institutional environments.

### 4.2. Higher Education Digitalization and Leadership

The result of mixing digital transformation and leadership within a higher education institutional context often results in a sub-research field known as digital leadership [164,165]. The practice of effective leadership within a digitally transformed higher education institutional field is urgently needed in order for institutional members and representatives to adapt to continuously changing demands and opportunities. Yet, there is little evidence of how leadership must be (re)defined to be fully operational for both administrative representatives (as administrative and teaching staff) and students (learners) together. Moreover, leadership is the core element for gender, racial, ethnic, religious and age diversity and inclusion within a highly institutionalized context. Whichever the design assessment, both transformational and transactional leadership styles are capable of creating a viable digital transformation organizational/institutional working platform by enhancing digitalization processes and mediating its implementation processes. The higher education digital environment is one of the pressing problems specific to the education sector. Across the majority of higher education institutions, the digital divide in education is reduced, since participants have access to high-speed internet and a large number of fixed and portable digital devices. The openness and accessibility specific to higher education institutions and their digital transformation processes are directly mediated through leadership processes specific to management and teaching staff, all reflecting on the student’s participation in the process. Within the current research, we found that higher education digital transformation is highly related to leadership processes, but surprisingly, at the core of digital processes is not the transformational leadership style, but the two components specific to transactional endeavors. Higher education digitalization and its components are partly explained by the two leadership components, confirming once again the utility of the study, identifying areas for institutional management to invest in and room for improvement. Starting from the internal digital transformation and leadership configuration of a larger array of universities, further developments should include the attributes, indicators and specific metrics to be fairly assessed and analyzed. For this reason, national leadership and higher education digitalization initiatives should be first provided with results in order to be subject to funding and development that will autonomously cultivate and improve their digital and leadership competencies.

### 4.3. Higher Education Digitalization and Students’ (Work) Engagement

The motivation process resides within the accessibility to work/learning resources, thus enhancing the learning resource outcomes (as work engagement) and resulting in a healthier and more productive organizational environment [166]. Starting from the idea that a (digitally) satisfied student is considered to be an asset for each higher education institution [167], with their work providing good results, research studies have shown that learners’ work engagement and productivity become higher as their satisfaction increases. The current results show a positive connection between students’ learning engagement and the institutional degree of digital transformation; even more, one specific component of students’ engagement in their studies has a larger impact on the institution’s digital transformation, namely, SS, which has been proven by the literature to also be a positive mediator between leadership and work (learning) engagement [168]. Further developments should also refer to particular digital transformation uses of practices specific to global learning platforms and intelligent learning connections that might include students, academic and management staff together, processes and data specific to the Internet of Things.

### 4.4. Theoretical Implications

The current study’s contributions to the literature include the first examination of the six proposed higher educational digitalization components, out of which only five could be validated. As the literature suggests [169], contextual studies have the benefit of bringing research closer to institutional and individual members’ and peers’ needs and realities; the respondents in the current study were selected only if they were actively enrolled students, with this fact resulting in a reduced risk of recall bias. For this reason, the measured digital transformation degree of the higher education institution is considered to have accurate results, compared to the situation of measuring recollections of what happened at a specific point in time.

It is clear from the previous observations that the current research is among the few of its kind to address the digital transformation degree of a higher education environment in the context of two other variables, leadership (transformational and transactional) and work (study) engagement. The current findings emphasize transactional leadership and one component of students’ (work) engagement, namely, Social Support, to be the best predictors of higher education digitalization; different literature results [170] suggest that, for common organizational contexts, CR is a good stimulator of work engagement; a contrary opinion [171] argues the idea that transactional leadership does not have the ability to influence individuals’ work engagement. From a theoretical standpoint, transformational leadership has the power to augment the role of the transactional leadership style. The literature has largely suggested that transformational and transactional behaviors result in different outcomes in a sustainable organizational context [172]; it appears that the current study supports this presumption, since the traditionally expected transformational and transactional leadership roles have reversed. It is worth mentioning that future developments might include the study of higher education institutions’ digital transformation in relation to all three components specific to transactional leadership, thus including, besides MBE active, the MBE passive component.

Since students are not provided with information sources for institutional behavior in the context of (re)acting when defending their interests, the current two-style leadership assessment for the same institutional context may prove to be useful in allowing a deeper student understanding of the advances of their institution in regard to digital transformation enhancements and managerial decisions in regard with their curriculum. It is important at this point for students not to trust the information provided by external peers, thus affecting their learning (work) engagement. The results show that there is a non-linear TL–student engagement relationship, enforcing once more the traditionally acclaimed organizational transformational leadership style.

Further, the importance of higher education digital transformation’s practical applicability for both learners and teaching staff has the ability to lead to higher education academic performance, promoting the further development of students when integrating and performing within an exterior public and/or private working environment.

### 4.5. Practical Implications

From a practical standpoint, the current manuscript shows the importance of enforcing a specific and accurate digital transformation assessment instrument, which could be further used for measuring investment trajectory and performance, within a higher education organizational environment.

Further, assessing the connection between the digital transformation level and students’ engagement in learning processes offers important information for future institutional developments that could result not only in improved academic performance but also in better organizational (institutional) student retention.

The current framework could not miss the assessment of leadership styles and performance, as perceived by learners; referring to leadership, when assessing the questionnaire, students were asked to refer to their teachers, and thus, the current research provides a final institutional characterization of the teaching–learning processes specific to the higher education environment. The results show an important turnover from transformational to transactional leadership, namely, CR and MBEA, which were rated as higher in importance when about the context created by higher education digital transformation and students’ learning engagement. As a contribution, the current results offer important insights for higher education institutional management to invest in and cultivate a transactional leadership style, thus increasing the communication level of a digitally transformed higher institution to the users of the implemented changes, namely, the enrolled students. According to previous research, leaders perform well under institutional umbrella training [173,174,175]; therefore, according to the results of the current study, transactional leadership should be further enhanced. It is important to also acknowledge that results show a deep connection between transformational leadership and students’ learning engagement; for this reason, the duality of the analyzed leadership styles and the results create an institutional map that clearly connects digital transformation investments acquired with the transactional leadership style and students’ learning engagement with transformational leadership.

Ultimately, it is important to relate practical higher institutions’ digital transformation to the teaching–learning processes that translate to the learner’s engagement, ultimately resulting in higher university rankings for academic and financial purposes, but also in successful students’ insertion into the labor market, thus increasing their chances for organizational development and performance across industries.

It is important to stress the importance of transforming and adapting leadership tools and techniques to the digital environment; as the current research shows, despite the increasing trendline suggesting that transformational leadership is the ultimate solution for creating and enhancing an enriched organizational culture, and thus creating added value and innovativeness, within a socially and digitally awake organizational atmosphere, transactional leadership, through its main components (and here, we actively depict contingent reward), seems to be gaining in importance, thus favoring a rift in the common institutional beliefs related to leadership. The organizational need to adapt to such changes becomes imperative, especially for international organizational activities, where the stringent need for innovation and the pressure to adapt internal behaviors to a continuous process of organizational change have become a never-ending process. In light of this, leadership within public and/or private universities cannot be regarded as just another set of channels and platforms but, given its latest digital developments, should be perfectly adapted so as to result in a new self-sustaining tool for assessing online/digital organizational activity.

According to modern pedagogies mainly centered on students, (higher education) organizational environments are spaces that favor a framework where students acquire learning attitudes and behaviors for transforming into leaders. This informal organizational platform, where students are followers of teachers seen as leaders and enhance individual leadership behaviors, does not contradict modern pedagogies that are centered on students, since these teaching philosophies and traditions create the exact aspiration for followers to pursue their mentor(s)’s path and even overcome their experience and performance.

### 4.6. Study Limitations

Setting aside the clear strengths of the current research that one can rely upon, its limitations must also be taken into consideration. According to the literature [115], the use of qualitative assessment tools has the potential to increase the bias risk, which might be reflected in the variance. Contrary to the current opinion [176,177], common methods are unlikely to be affected by bias, a situation in which the current research is found, since the proposed construct shows relationships that are qualified as moderate. Moreover, the risk of potential bias has also been reduced by the anonymity of the respondents [115].

Another limitation might reside within the use of only two out of three transactional leadership components (excluding MBE passive). Since the literature provides previous results that show that the leadership components do not always relate to the study’s organizational environment (such as a higher education institutional environment), one can count on the importance of the results that relate to an inverted importance scale that particularizes the leadership style practice. Since MBE passive is mainly used when the environment under assessment presents difficulties for monitoring mistakes that involve large effects [178], the authors decided not to include MBE passive, as respondents (actively enrolled students) would not have access to such information, therefore producing biased results. Additionally, considering the institutional tendency for group (departmental/field/faculty) work, the authors assumed the predominant use of transactional leadership with its two components as CR and MBEA. Furthermore, MBE active and passive have been reported to produce similar results in response to the timing specific to the leadership style practice. For future research, the authors suggest applying the two leadership styles in their complete forms, as previously suggested [109,172,179] (thus also including MBE passive).

For the sample and context sizes, future research could include a larger number of higher education contexts by also including a representative number of respondents for each faculty and/or field of study.

The number of respondents might also affect the generalizability of the findings; thus, the results are in line with the hypothesis derived from the specific field’s literature context; despite the current approach, it is recommended that, before higher education institutions apply the results, they replicate the research with larger respondent pools.

Performing the current research and focusing on the reliability and validity of the higher education digital transformation assessment tool in relation to the academic practice of two leadership styles could ultimately result in a decrease/increase in enrolled students’ work (learning) engagement. The results provide proof of the stability of the construct design, thus providing a reliable and operational working platform for further institutional developments, including further analysis. Naturally, in order to be applied, future higher education institutional endeavors should apply to specific work (learning) frameworks, thus considering digital transformation’s economic and/or social advancements, fields of investments and/or study, the development of the labor market and its specific demands, and technological advances that, due to their development speed, may be problematic for institutional investments in acquiring a fully digitally transformed higher education institutional framework.

For this reason, a question might arise regarding the nature of the digital transformation circumstances specific to a higher education institution that might lead to enhanced student learning engagement. Prior literature findings [180,181] suggest that the answer resides within institutional leadership practices, since leaders have a lower influence on the work/study engagement of followers when not provided with institutional (online) learning (re)sources. Further, MBE active has been reported to increase its efficiency and effectiveness when applied to high-risk environments, thus resulting in sustainable results at institutional levels.

## 5. Conclusions

The contributions of the current study to the literature include that, to our knowledge, it is the first attempt to provide and validate an assessment tool that analyzes the degree of digital transformation specific to a higher education institution. Furthermore, the novelty of the digital transformation assessment tool in regard to the practice of the two leadership styles (transformational and transactional) is revealed through its results, which show that transactional leadership components are preferred to the transformational leadership style within a digitally transformed higher education institutional framework. Moreover, we assessed the practice of leadership styles in the digital transformation context in relation to students’ (learners’) engagement when performing their required tasks and study activities. Additionally, we examined not only the linear correlations among the referred variables but also the quadratic (thus non-linear) effects. Ultimately, the specific indirect effect sizes were measured, with results showing the existence of small to moderate effects for several of the measured variables, such as EXP, ATI and LDS—higher education digitalization—TL. To conclude, the current study provides a sustainable working platform that benefits higher education institutions’ assessment of their past/current/future developments of digital transformation in relation to the necessary leadership teaching–learning practices that would ultimately result in students’ learning engagement, thus providing such institutions with insights for future financial and social endeavors and students with higher chances of successful labor market insertion and performance.

## Figures and Tables

**Figure 1 ijerph-20-05124-f001:**
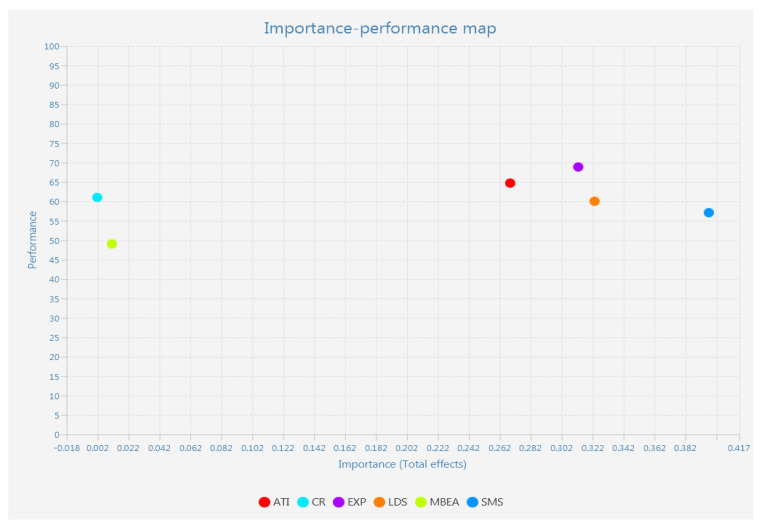
Higher education digitalization Importance–Performance Map. Source: authors’ calculation with SmartPLS (v. 4.0.0) software.

**Figure 2 ijerph-20-05124-f002:**
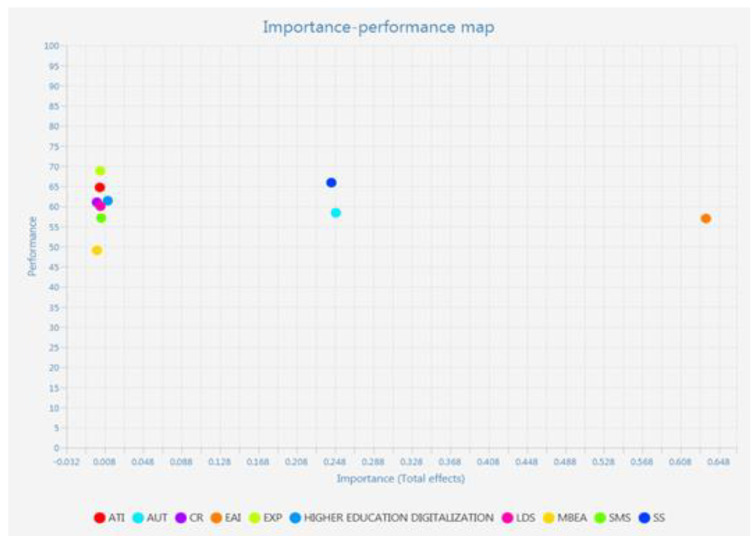
Students’ engagement Importance–Performance Map. Source: authors’ calculation with SmartPLS (v. 4.0.0) software.

**Figure 3 ijerph-20-05124-f003:**
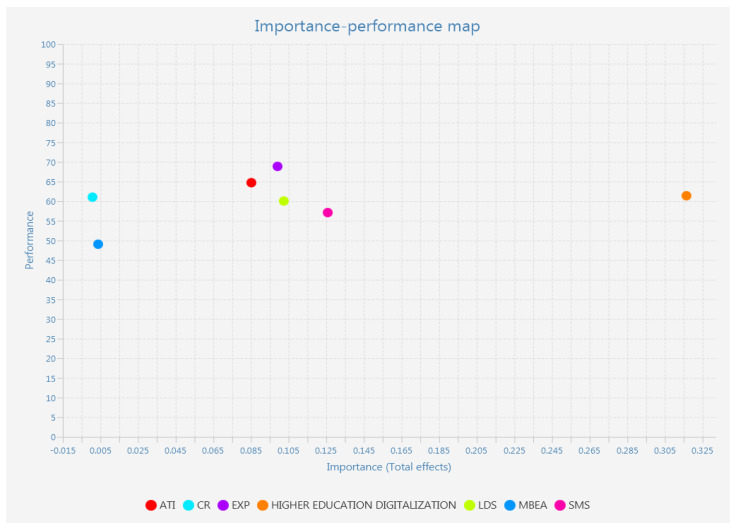
Transformational leadership Importance–Performance Map. Source: authors’ calculation with SmartPLS (v. 4.0.0) software.

**Figure 4 ijerph-20-05124-f004:**
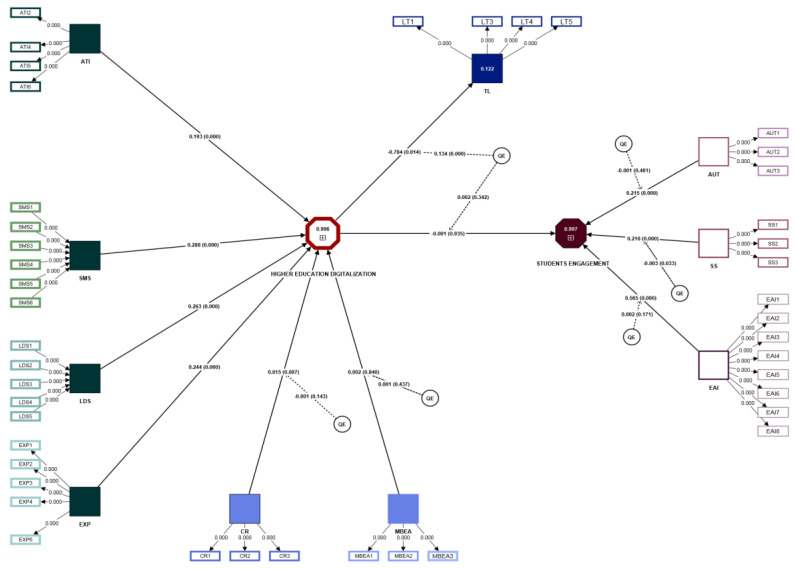
Higher education digitalization model’s curvilinear quadratic effects. Source: authors’ calculation with SmartPLS (v. 4.0.0) software.

**Figure 5 ijerph-20-05124-f005:**
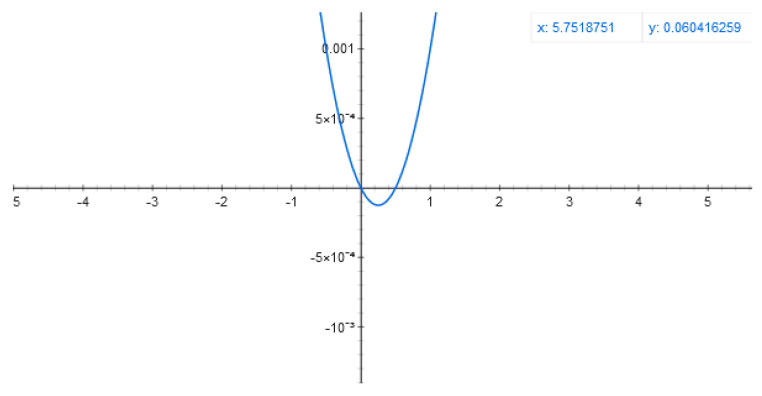
Higher education digitalization–TL curvilinear quadratic effects—model plot. Source: authors’ calculation with SmartPLS (v. 4.0.0) software.

**Figure 6 ijerph-20-05124-f006:**
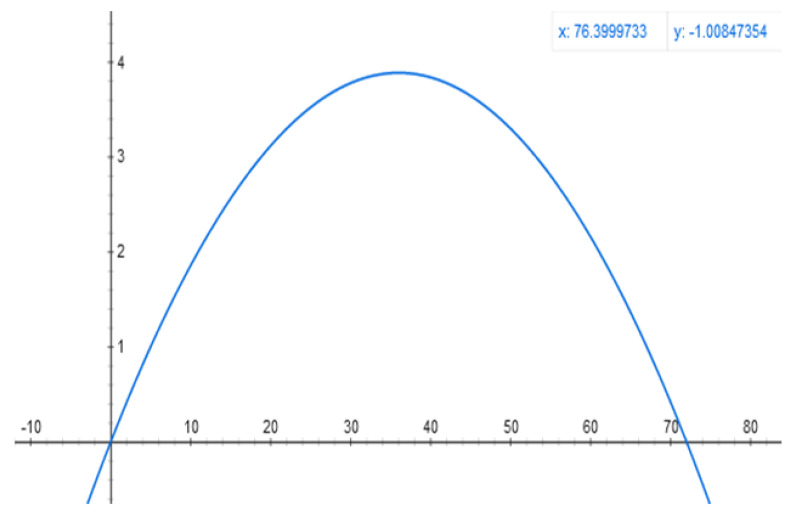
SS–students’ engagement curvilinear quadratic effects—model plot. Source: authors’ calculation with SmartPLS (v. 4.0.0) software.

**Table 1 ijerph-20-05124-t001:** Initial and final construct reliability and validity.

	Cronbach’s Alpha (*F)	Cronbach’s Alpha (*I)	Composite Reliability (rho_a) (*F)	Composite Reliability (rho_c) (*F)	Average Variance Extracted (AVE) (*F)	Average Variance Extracted (AVE) (*I)
ATI	0.695	0.695	0.691	0.814	0.525	0.524
AUT	0.826	0.826	0.828	0.896	0.741	0.741
CR	0.82	0.82	0.833	0.893	0.735	0.735
EAI	0.919	0.919	0.922	0.934	0.64	0.64
EXP	0.704	0.704	0.73	0.811	0.469	0.469
HIGHER EDUCATION DIGITALIZATION	0.848	0.856	0.852	0.874	0.26	0.236
LDS	0.649	0.649	0.652	0.781	0.419	0.42
MBEA	0.758	0.767	0.759	0.892	0.805	0.683
SMS	0.739	0.739	0.744	0.821	0.434	0.434
SS	0.814	0.814	0.817	0.89	0.73	0.73
STUDENTS’ ENGAGEMENT	0.93	0.93	0.934	0.939	0.511	0.511
TL	0.879	0.879	0.891	0.911	0.671	0.671
EQP		0.442				0.369

* F = final; I = initial; source: authors’ calculation with SmartPLS (v. 4.0.0) software.

**Table 2 ijerph-20-05124-t002:** Construct path coefficients.

	Path Coefficients
Ati → higher education digitalization	0.273
Aut → students’ engagement	0.249
Eai → students’ engagement	0.635
Exp → higher education digitalization	0.316
Higher education digitalization → cr	0.286
Higher education digitalization → mbea	0.275
Higher education digitalization → students’ engagement	0.011
Higher education digitalization → tl	0.317
Lds → higher education digitalization	0.329
Sms → higher education digitalization	0.399
Ss → students’ engagement	0.244

Source: authors’ calculation with SmartPLS (v. 4.0.0) software.

**Table 3 ijerph-20-05124-t003:** Construct R-square values.

	R-Square	R-Square Adjusted
CR	0.082	0.08
HIGHER EDUCATION DIGITALIZATION	1	1
MBEA	0.076	0.075
STUDENTS’ ENGAGEMENT	0.997	0.997
TL	0.1	0.099

Source: authors’ calculation with SmartPLS (v. 4.0.0) software.

**Table 4 ijerph-20-05124-t004:** FIMIX segment sizes.

	S 1	S 2	S 3	S 4	S 5	S 6	S 7	S 8	S 9	S 10	S 11	S 12	S 13
%	0.304	0.129	0.112	0.08	0.072	0.06	0.052	0.046	0.041	0.039	0.035	0.021	0.01

Source: authors’ calculation with SmartPLS (v. 4.0.0) software.

**Table 5 ijerph-20-05124-t005:** FIMIX results.

AIC3 (modified AIC with Factor 3)	−2562.05	−2654.47	−2663.208	−2655.27	−2640.62	−2616.52	−2604.93	−2571.85	−2549.46	−2522.19	−2505.28	−2467.87	−2453.619
CAIC (consistent AIC)	−2495.53	−2516.69	−2454.159	−2374.96	−2289.04	−2193.67	−2110.82	−2006.47	−1912.81	−1814.28	−1726.1	−1617.42	−1531.905
EN	0	0.313	0.355	0.377	0.403	0.399	0.46	0.501	0.5	0.499	0.539	0.521	0.524
SUMMED FIT	−5057.58	−5171.15	−5117.367	−5030.23	−4929.65	−4810.18	−4715.74	−4578.33	−4462.28	−4336.47	−4231.37	−4085.28	−3985.524

Source: authors’ calculation with SmartPLS (v. 4.0.0) software.

**Table 6 ijerph-20-05124-t006:** Specific indirect effect sizes.

	Original Sample	Sample Mean (M)	Standard Deviation (STDEV)	T Statistics (|O/STDEV|)	*p*-Values	Indirect Effect Size
Sms → higher education digitalization → tl	0.157	0.157	0.019	8.469	0	Less than a small effect
Exp → higher education digitalization → tl	0.146	0.146	0.017	8.602	0	Small effect
Ati → higher education digitalization → tl	0.123	0.123	0.016	7.828	0	Small effect
Lds → higher education digitalization → tl	0.122	0.122	0.015	7.96	0	Small effect
Cr → higher education digitalization → tl	0.002	0.002	0.007	0.232	0.817	Less than a small effect
Mbea → higher education digitalization → tl	0	0.001	0.004	0.117	0.907	Less than a small effect
Qe (mbea) → higher education digitalization → students’ engagement	0	0	0	0.489	0.625	Less than a small effect
Qe (cr) → higher education digitalization → students’ engagement	0	0	0.001	0.104	0.917	Less than a small effect
Mbea → higher education digitalization → students’ engagement	0	0	0	0.108	0.914	Less than a small effect
Cr → higher education digitalization → students’ engagement	0	0	0	0.222	0.824	Less than a small effect
Qe (cr) → higher education digitalization → tl	−0.001	−0.001	0.012	0.107	0.915	Less than a small effect
Lds → higher education digitalization → students’ engagement	−0.005	−0.005	0.002	2.631	0.009	Less than a small effect
Ati → higher education digitalization → students’ engagement	−0.005	−0.005	0.002	2.72	0.007	Less than a small effect
Exp → higher education digitalization → students’ engagement	−0.006	−0.006	0.002	2.65	0.008	Less than a small effect
Qe (mbea) → higher education digitalization → tl	−0.006	−0.006	0.012	0.513	0.608	Less than a small effect
Sms → higher education digitalization → students’ engagement	−0.006	−0.006	0.002	2.671	0.008	Less than a small effect

Source: authors’ calculation with SmartPLS (v. 4.0.0) software.

## Data Availability

Not applicable.

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
