# Peer review of "The Role of Leadership and Digital Transformation in Higher Education Students’ Work Engagement"

_ijerph, 2023, doi:10.3390/ijerph20065124_

Round 1

Reviewer 1 Report

Thank you for providing your manuscript for a review. It is a consistent paper, with a well-reported analysis. The results are well depicted in relation to the hypotheses. The assessment tool that the authors prepared should certainly be appreciated. 

Few recommendations in order to improve the general clarity of the paper:

66-68: I wonder if this affirmation is just too optimistic; many factors could intervene and destabilize the effects of students’ engagement.

69-75: The formulation could be improved – there are many ideas that overlap and are hard to follow, mainly because they are in Introduction, where the context of understanding is still limited. 

The evolutionary paradigm of leadership is a very interesting framework; I would suggest that this paradigm be briefly explained and referenced in the Introduction. Also, its entanglements with the digital are not clear for a common reader of the paper and they would deserve an explanation. (see lines 47-49).

The authors provide at the end of the introductory section the needed description of the paper structure, but it could be a little better presented, taken into consideration its relevance and usefulness for readers (a project is mentioned– we do not know if this word represents merely a substitute for the current paper or it is a reference to a certain research project; also, some minor linguistic inaccuracies that have to be corrected obscure the clarity of this part.). 

Please introduce the abbreviation between brackets in the text, after the terms or expressions abbreviated, before using it alone. Please check the repetitions of a term in the same phrase and reformulate them for a better understanding (as in lines 107-109 etc.)

H1 is somehow abruptly introduced; the reader could not understand why these specific six attributes were selected. I would suggest the insertion of few more arguments of this particular selection and of their importance for the subject.

In general, it is not self-evident if every teacher is considered a leader and the leadership is considered also at this level (in the classroom) or the leadership is seen only at the university management level? Please clarify the levels of discussion. 

When the authors talk about students, it seems that work is equated with learning and also there are references to work as job, workforce, future work after graduation etc. Please provide explanation for the situations where what is meant by “work” is attributed to learning. There are differences between work and learning or they really juxtapose perfectly? I suggest a brief clarification of this matter especially because the term is part of the title.   

For an in-depth understanding of the complex subject analyzed, it would be a plus if the authors would try to answer questions like: 

Is leadership so significantly shaped by digital environment, or new media represent just another set of channels and platforms that can be used in the development and transmission of traditional leadership? In this vein, maybe a few examples of digital leadership activities in universities could optimize the understanding for every reader.  

If the teachers are the leaders and the students are the followers, is this situation somehow in contradiction with the modern pedagogies that are centered on students and also create specific educational spaces where students themselves could be leaders? If the digital skills are to be acquired also by teachers, how could they be digital leaders, in the context where digital natives are presumably more advanced?

Author Response

Dear Reviewer,

Thank you very much for your time and input on our manuscript. We tried to answer all the suggestions the better we could.

We reduced the optimism on the 66-68 lines; indeed, it was a too great generalization.

We improved the formulation on lines 69-75.

We added information and explanations in regard to the evolutionary paradigm of leadership within the Introduction section and clarified lines 47-49.

We introduced at the end of the introductory section a paper structure.

We proofed the text for abbreviations, misspelling, punctuation and repetitions.

We explained better the context in regard with H1.

We explained the paradigm leader-follower- teacher-student.

We explained the implications for work- learning process, similitudes and differentiation.

We discussed on the basis of the last two questions in regard to development and transmission of traditional leadership and the effect of modern pedagogies on education.

We thank you very much once again for your time, appreciation and input.

Best regards,

Reviewer 2 Report

This paper is really interesting, and the contribution of the study is clear. This study examines o examine 16 how higher education institutions should apply different leadership styles within digitally trans- 17 formed contexts, as to increase the student’s learning engagement and reduce risk of failure on their 18 future developments within international and national labor markets.

In the introduction, main idea is clear where authors explain and detail their contribution and the objectives of their research. But authors could be considered to write two part of this part. One of then is related to Introduction and another part is related to Review Literature. It is also suggested that at the end of the introduction you include a paragraph summarising the sections to be covered in the manuscript. This would help readers to understand and have a more general overview of the work. Author make good justifications. Also, they have included more current references, but it would be interesting to include some references from the journal itself. It is suggested that they be included:

Aránega, A. Y., Montesinos, C. G., & del Val Núñez, M. T. (2023). Towards an entrepreneurial leadership based on kindness in a digital age. Journal of Business Research159, 113747.

Aránega, A. Y., Sánchez, R. C., & Pérez, C. G. (2019). Mindfulness' effects on undergraduates' perception of self-knowledge and stress levels. Journal of Business Research101, 441-446.

Bartsch, S., Weber, E., Büttgen, M., & Huber, A. (2020). Leadership matters in crisis-induced digital transformation: how to lead service employees effectively during the COVID-19 pandemic. Journal of Service Management32(1), 71-85.

Cortellazzo, L., Bruni, E., & Zampieri, R. (2019). The role of leadership in a digitalized world: A review. Frontiers in psychology10, 1938.

Shin, J., Mollah, M. A., & Choi, J. (2023). Sustainability and Organizational Performance in South Korea: The Effect of Digital Leadership on Digital Culture and Employees’ Digital Capabilities. Sustainability15(3), 2027.

The methodology is clearly presented, helping readers to understand better main objectives. The author mention the hypotheses in a clear way. The contribution of the figures and stages are adequate, they help to better understand the methodology. But they need to be improved, so that the tables keep the same format.

The data analysis is well justified. The author have taken into account the above suggestions, where they have included data analysis in a separate section. This is important, as the quantitative part is now more clearly identified.

It is suggested that a separate Discussion section be added to the literature review. In addition, apart from focusing on the data, be related to the literature review.

Finally, authors are advised to have any spelling mistakes checked by a professional copyediting service.

Author Response

Dear Reviewer,

Thank you very much for your time and input on our manuscript. We tried to answer all the suggestions the better we could.

We modified the Introduction section as suggested and added information in regard to the structure of the manuscript.

We added the relevant literature that you provided. Thank you very much for your kind suggestions.

We worked on the format of the tables, as much as possible given the Journal requirements in this regard.

We added a separate Discussion section to the literature review.

We proofed the text for abbreviations, misspelling, punctuation and repetitions.

We thank you very much once again for your time, appreciation and input.

Best regards,